# EMA-YOLO: A Novel Target-Detection Algorithm for Immature Yellow Peach Based on YOLOv8

**DOI:** 10.3390/s24123783

**Published:** 2024-06-11

**Authors:** Dandan Xu, Hao Xiong, Yue Liao, Hongruo Wang, Zhizhang Yuan, Hua Yin

**Affiliations:** 1School of Software, Jiangxi Agricultural University, Nanchang 330045, China; 15070809158@163.com (D.X.); 18279980023@163.com (Y.L.); wanghongruo0411@163.com (H.W.); yuanzz88888@163.com (Z.Y.); 2School of Software, Jiangxi Normal University, Nanchang 330045, China; 18379976616@163.com

**Keywords:** target detection, attention module, yellow peach, YOLOv8

## Abstract

Accurate determination of the number and location of immature small yellow peaches is crucial for bagging, thinning, and estimating yield in modern orchards. However, traditional methods have faced challenges in accurately distinguishing immature yellow peaches due to their resemblance to leaves and susceptibility to variations in shooting angles and distance. To address these issues, we proposed an improved target-detection model (EMA-YOLO) based on YOLOv8. Firstly, the sample space was enhanced algorithmically to improve the diversity of samples. Secondly, an EMA attention-mechanism module was introduced to encode global information; this module could further aggregate pixel-level features through dimensional interaction and strengthen small-target-detection capability by incorporating a 160 × 160 detection head. Finally, EIoU was utilized as a loss function to reduce the incidence of missed detections and false detections of the target small yellow peaches under the condition of high density of yellow peaches. Experimental results show that compared with the original YOLOv8n model, the EMA-YOLO model improves mAP by 4.2%, Furthermore, compared with SDD, Objectbox, YOLOv5n, and YOLOv7n, this model’s mAP was improved by 30.1%, 14.2%,15.6%, and 7.2%, respectively. In addition, the EMA-YOLO model achieved good results under different conditions of illumination and shooting distance and significantly reduced the number of missed detections. Therefore, this method can provide technical support for smart management of yellow-peach orchards.

## 1. Introduction

Yellow peaches are favored by consumers due to their nutrient composition and deliciousness [1]. As their quality of life improves, more and more people are becoming concerned about their health, and the focus of the demand for yellow peaches has shifted from quantity to quality. As a result, more and more farmers utilize various new technologies to manage modern orchards, and accurately counting the number of immature yellow peaches is becoming crucial [2]. In a practical scenario, by utilizing statistical data regarding immature yellow peaches, growers can optimize decision-making about purchasing bagging materials and employing workers [3]. Furthermore, such data also can significantly enhance both the yield and quality of yellow peaches while optimizing orchard management.

However, due to the complexity of the orchard environment, manual estimation of the number of peaches is the primary method used [4]. This method is obviously inaccurate, low-efficiency and high-cost, which makes it challenging to adapt it to the intelligent management of large yellow-peach orchards. In response to these challenges, researchers have conducted in-depth research. Particularly with the continuous advancement of computer vision technology, an increasing number of researchers have begun utilizing visual-detection technology for in situ fruit counting [5]. The series of YOLO algorithms are particularly favored by researchers and engineers due to their rapid detection speed [6]. In 2021, Song et al. [7] used the dense connection mechanism of DenseNet to replace the last three down-sampling layers of the Darknet53 feature-extraction network in the YOLOv3 network in order to enhance feature propagation and achieve repeated use of features. As a result, the mean accuracy of green citrus identification reached 80.98%. In 2022, Hao et al. [8] introduced a hybrid data-enhancement method for detection of green walnuts and replaced the backbone of the YOLOv3 algorithm with MobileNet-v3, achieving an mean accuracy of 94.52%. Song et al. [9] improved YOLOv5 to allow it to recognize oil fruits in natural scenes, achieving an mean accuracy of 98.71%. Zhang et al. [10] added a transformer module with attention mechanism and resulted in a 3.77% increase in mAP. Lv et al. [11] added a stripe attention module to the backbone network of YOLOv5, enabling the model to pay more attention to the stripe-shaped sleeves of citrus fruits and branches. At the same time, they provided a semi-supervised method as a student model in the classroom, enabling the target-detection algorithm to use unlabeled samples and thus improving the performance of the model and reducing its reliance on labeled samples. The mean accuracy of the improved algorithm in the detection of sleeve citrus and branches reached 77.4% and 53.5%, respectively. Xie et al. [12] added an attention module to YOLOv5 and modified the loss function by including a small-target-detection layer, resulting in a 12.9% improvement in mean accuracy for a litchi dataset. Zhang et al. [13] developed a fruit-counting algorithm based on YOLOx that utilizes sample enhancement of specific scenes. Tang [14] successfully developed YOLOv7-Plum to detect plums in natural environments. In 2023, our team proposed an improved scheme based on YOLOv7 for yellow-peach detection; this innovation achieved an improved detection mean accuracy of 80.4% because we incorporated a CA attention-mechanism module and modified the loss function [15].

The above studies demonstrated the effectiveness of the YOLO algorithm for fruit detection in orchards. Despite this, in immature peach orchards, detecting targets using the original YOLO network is challenging due to factors such as complex and varied backgrounds, small fruit sizes, and frequent occlusions. Thus, further improvements are needed to enhance its target-detection ability.

The YOLOv8 algorithm was put forward in January 2023 and established a new SOTA (state of the art) for the target-detection model. To further improve the recognition rate of immature small yellow peaches, a novel detection algorithm based on YOLOv8, called EMA-YOLO, is proposed. The main contributions of this study can be stated as follows:

① Introduction of the EMA (Efficient Multi-Scale Attention) attention-mechanism module to encode global information and further aggregate pixel-level features through dimensional interaction.

② Combination with the 160 × 160 scale detection head to enhance small-target-detection capabilities.

③ Employ EIoU (Efficient Intersection over Union) as the loss function to reduce the rates of missed detections or false detections of small target yellow peaches in dense environments.

These improvements are tailored to address the specific challenges posed by detecting immature small yellow peaches in natural environments.

## 2. Materials

### 2.1. Data Acquisition

The yellow-peach samples were gathered from a plantation base in Daping Village, Jinggangshan City, Jiangxi Province (Figure 1). The data-collection period was from April to May 2022 and 2023. To enhance the algorithm’s versatility and facilitate subsequent UAV (Unmanned Aerial Vehicle) operations, images were captured under different lighting conditions and at different shooting distances and angles using a mobile phone. Each image had a resolution of 4000 × 3000 pixels. To ensure result accuracy, blurry images were manually excluded, and the remaining images were evenly distributed across different scenes. A total of 1520 original images were obtained, of which 498 were long-distance images (more than 3 m), 506 were medium-distance images (1–3 m) and 516 were close-distance images (less than 1 m). The dataset is divided into a training set, a verification set and a test set at a ratio of 8:2.

The images were annotated by Labelimg. The annotation file was then stored in XML (extensible markup language) format. The file contained the vertex coordinates of the yellow-peach target rectangle and the number of labeled instances, totaling 36,872 instances.

### 2.2. Data Augmentation

Data augmentation is a technique widely used in deep learning to improve generalization ability and reduce the risk of overfitting [16,17,18,19]. In this experiment, we utilized a feature-enhancement-based data-augmentation algorithm that can improve the visual impact of images [20]. The data-augmentation methods employed in this study included brightness adjustments, random rotation and inversion. The details are shown in Figure 2. As a result, 9120 images were obtained.

To obtain reliable results, the randomness of dataset partitioning was eliminated. In this experiment, the test was carried out using five-fold cross-validation. At the training process, one sub-data set was taken as the test set and the other four were used as the training set. The whole training process was repeated five times. Due to the slight differences in the performance of the model on different training sets, the test results also fluctuated slightly. Finally, the average was taken as the final result. This method ensures that each sample is trained and tested, thus reducing the generalization error.

## 3. Method

### 3.1. YOLOv8

YOLOv8 is a single-stage target-detection algorithm proposed by the Ultralytics company in 2023. It includes five versions: YOLOv8n, YOLOv8s, YOLOv8m, YOLOv8l and YOLOv8x. The number of model parameters and computational complexity increase with the depth and width of the model. Users can select the network structure based on their specific application scenarios. Notably, YOLOv8-n is specifically designed to be used on embedded devices while maintaining detection speeds and accuracy [21]. To apply our algorithm on the mobile device, we choose the network structure of YOLOv8n. The structure diagram of YOLOv8n is depicted in Figure 3. The entire network comprises a backbone for feature extraction, a neck network, and a detection head for feature fusion. The backbone section adopts a cross-stage local network structure to reduce computational load and improve gradient strength. Additionally, it incorporates a spatial pyramid pooling module to improve spatial feature extraction. The head section employs the current mainstream decoupled head to effectively decrease parameter count and computational complexity while enhancing the model’s generalization ability and robustness. YOLOv8 also departs from previous designs in the YOLO series that used anchor-based methods to predict anchor box position and size; instead, it utilizes an anchor-free detection method to directly predict target center points and width/height ratios. This reduction in anchor boxes further improves the model’s detection speed and accuracy.

### 3.2. EMA-YOLO

The detection of immature yellow peaches presents several challenges, including small-target detection and natural scene occlusion. To address these issues, we have proposed the EMA-YOLO model, which incorporates an EMA attention-mechanism module into the YOLOv8n framework. This enhancement includes an additional 160 × 160 small detection head and utilizes EIoU (Efficient Intersection over Union) as a loss function to achieve lightweight and accurate model improvements. The network-structure diagram for EMA-YOLO is depicted in Figure 4, with red boxes highlighting the improvements in the model.

#### 3.2.1. EMA Attention Mechanism

Immature yellow peaches can easily be confused with leaves due to their color and small size. This poses a challenge for the traditional YOLOv8 detection model. However, it was noted that yellow peaches typically grow on the main stem and tend to aggregate. Thanks to this feature, an attention-mechanism module is a potentially useful method for enhancing performance on integration into the original YOLOv8 network.

Currently, there are various commonly used attention-mechanism modules available; among them is the Efficient Multi-Scale Attention [22] module, which was designed for learning across space without dimensionality reduction. EMA utilizes a grouping structure and employs cross-space learning methods to establish short- and long-term dependencies through a multi-scale parallel subnetwork design. It retains information from each channel through cross-dimensional interactions while grouping channel dimensions into multiple sub-features.

The spatial semantic features are evenly distributed within each feature map, effectively establishing short- and long-term dependence relations through feature grouping and the multi-scale structure of EMA, ultimately contributing to improved detector performance while reducing parameter requirements and computational overhead. The structure of the EMA attention-mechanism module is shown in Figure 5.

The EMA attention mechanism preserves precise location information by modeling long-term dependencies. The EMA attention-mechanism model includes long-term dependencies and preserves precise location information. Because context information is integrated at various scales, neural networks are empowered to produce more accurate pixel-level attention for feature maps. Additionally, the parallelization of convolution kernels provides a more robust structure that utilizes cross-space learning methods to handle short- and long-term dependencies. This is achieved by employing 3 × 3 convolution and 1 × 1 convolution in parallel to incorporate more contextual information into the intermediate feature graph. EMA-YOLO leverages the output of the EMA attention-mechanism module combined with channel information and context data to differentiate between occlusive yellow peaches at different scales, effectively distinguishing them from the leaves that are similar in color to the unripe yellow peaches.

#### 3.2.2. Incorporate the Small-Object Detection Head

The size of the peaches changes with shooting distance, which is a common phenomenon in practical orchard management. This leads to significant scale-based differences in the detection of peaches. In images taken from a long distance, a yellow peach occupies less than 1% of the image size, resulting in a potential loss of feature information. Zhu et al. [23] addressed this issue by adding a transformer detection head to their model, achieving successful results in detecting densely arranged objects with dramatic scale changes during high-speed and low-altitude flight. Inspired by this study, we add a 160 × 160 small-target detection head to enhance sensitivity in detection of smaller targets. EMA-YOLO initially extracts features from the sixth layer of the backbone network and utilizes concat splicing to fuse shallow features extracted from the neck network structure with context information extracted from an EAM attention-mechanism module. Finally, we used the output fourth detection head at layer 21 as our small-object detection head.

While this improvement slightly increases computational load, it significantly enhances EMA-YOLO’s ability to detect small objects by capturing more detailed feature information and effectively reducing false detections and missed detections of yellow peaches across different scales. The improved detection layer is shown in Figure 6.

#### 3.2.3. EIoU Loss Function

The original YOLOv8n algorithm uses the prediction box loss function CIoU (Complete Intersection over Union) [24], as shown in Equation (1).
(1)$$L_{CIOU}=1-IOU+\frac{\rho^{2}(b,b^{gt})}{c^{2}}+\alpha v$$
(2)$$\alpha =\frac{v}{(1-IOU)+v}$$
(3)$$v=\frac{4}{\pi^{2}}(\arctan\frac{w^{gt}}{h^{gt}}-\arctan\frac{w}{h}{)}^{2}$$

*IoU* (intersection over union) represent the ratio of the intersection and union between the bounding box and the real box; b and $b^{gt}$ represent the center points of the bounding box and the real box; $\rho^{2}$ is the Euclidean distance between two points. α is the penalty factor; v represents the similarity of aspect ratio; c is the diagonal length of the minimum closure region; w and h represent the width and height of the boundary frame; and $w^{gt}$ and $h^{gt}$ represent the width and height of the real box, respectively.

The above formula demonstrates that CIoU (complete intersection over union) incorporates the aspect ratio of the bounding box as a penalty term into the bounding-box loss function, which can enhance the convergence speed of the regression process for the prediction box to some extent. However, when there is a linear relationship between the width and height of the prediction box and those of the real box, simultaneous adjustment of both dimensions in regression becomes impossible. As a result, the penalty term loses its effectiveness in describing the regression objective accurately, potentially leading to slow convergence and inaccurate regression. To address this issue, EIoU [25] is introduced in this paper, utilizing a loss-function expression that directly penalizes w and h in Equation (4), as shown below:(4)$$L_{EIoU}=L_{IoU}+L_{dis}+L_{asp}\;=1-IoU+\frac{\rho^{2}(b,b^{gt})}{c^{2}}+\frac{\rho^{2}(w,w^{gt})}{c_{w}^{2}}+\frac{\rho^{2}(h,h^{gt})}{c_{h}^{2}}$$
where $L_{IoU}$, $L_{dis}$ and $L_{asp}$ represent the losses for overlap, distance and width/height, respectively; IoU is the ratio of the intersection to the union between the predicted bounding box and the ground-truth bounding box. *c*, *c_w_* and *c_h_* denote the diagonal length, width and height of the minimum enclosing rectangle for two frames. $\rho (b,b^{gt})$ represents the distance between the center points of the predicted and ground-truth bounding boxes; $\rho (w,w^{gt})$ indicates the difference in width between the predicted and ground-truth bounding boxes; $\rho (h,h^{gt})$ signifies the difference between the predicted and ground-truth bounding boxes.

The EIoU loss function minimizes the discrepancy in width and height between the target and the anchor. It takes into consideration various factors such as overlap area and distance between center points, as well as real disparities in width, height and side length. This approach enables the model to prioritize high-quality anchor frames during regression, leading to faster convergence speed and improved regression prediction accuracy while maintaining strong anti-interference capability. The introduction of EIoU significantly enhances performance in object-detection tasks, particularly for yellow-peach detection in conditions of severe occlusion.

### 3.3. Experimental Parameters

The experimental environment configuration is as follows: the operating system of the experimental platform is Windows 10 (professional edition) with a Nvidia GeForce RTX 3090Ti graphics card and 24 GB of video memory (Santa Clara, CA, USA). The software setup includes CUDA 11.7, Python 3.8, and the PyTorch 1.7 deep-learning framework. The training process consists of 450 epochs, with a batch size of 4 for reading image data. The initial learning rate is set to 0.0001.

### 3.4. Evaluation Metrics

The main relevant metrics utilized in this experiment to assess the impact of the neural network model include precision, recall, F1 score [26], PR curve, and mean average precision. Precision indicates the proportion of correctly predicted positive samples out of all actual positive samples, while recall represents the proportion of correctly predicted positive samples out of all actual positive samples. The F1 score is a weighted average of precision and recall. Generally, a higher F1 score suggests a more stable and robust model. Mean average precision (mAP) measures the combined impact of both precision and recall across all n categories [27]; thus, mAP was selected as the primary measure for model evaluation in this study.

The metrics comprehensively evaluate the precision, recall, and F1 score for model validation. The above-mentioned metrics are calculated as follows:(5)$$p=\frac{TP}{TP+FP}$$
(6)$$R=\frac{TP}{TP+FN}$$
(7)$$F_{1}=\frac{2\times P\times R}{P+R}$$

*TP* stands for true cases (TP); *FP* stands for false positive cases (FP); *TN* stands for true negative cases (TN); *FN* stands for false negative cases (FN) [28].

## 4. Experimental Results and Analysis

### 4.1. Experimental Result

The experimental results demonstrate that the improved EMA-YOLO model achieved a precision (P) of 0.836 and a recall rate (R) of 0.744, with a corresponding F1 score of 0.787. The precision curve, recall curve, mAP curve and loss curve of YOLOv8 and EMA-YOLO were compared, as shown in Figure 7. It can be seen that the precision, recall and mAP have been improved, and the loss function converges faster. Some of the test results from the orchard are shown in Figure 8.

### 4.2. Ablation Experiment

Ablation experiments were conducted on EMA-YOLO to evaluate the impact of each improvement, and the results are shown in Table 1. The findings suggest that data augmentation resulted in a 1.6% improvement in mAP, demonstrating its effectiveness in expanding the sample space and improving detection accuracy through enhanced sample diversity. Additionally, integration of the EMA attention-mechanism module resulted in a 1.1% improvement in mAP, highlighting its ability to enhance feature extraction and overall network accuracy.

Furthermore, appending the detection head also resulted in an 0.9% improvement in mAP due to the improved suitability of the p2 detection head for small-object detection, thereby mitigating loss of small-object information as network depth increases.

Moreover, introducing the EIoU loss function resulted in an 0.6% improvement in model mAP. In addition, incorporation of FocalLoss effectively addressed sample imbalance within bounding-box regression tasks by prioritizing high-quality anchor boxes over those with minimal overlap with target boxes.

The results of ablation experiments demonstrate a significant improvement in the mean average precision (mAP) of the model. In this experiment, the shallow features extracted from the neck network structure were fused with the context information extracted from the EAM attention-mechanism module, and the mAP was increased by 2% on input into the small-target detection head. This improvement is mainly due to the fact that our homemade yellow-peach dataset contains a large number of heavily obscured yellow peaches. When EMA is combined with the small-target detection head, the network can detect more small targets and the accuracy increases. On this basis, the exclusion loss function was replaced to further improve the model, allowing it to accurately locate each yellow peach in dense environments and reduce the rates of missed and false under conditions of severe occlusion. The incorporation of data augmentation, utilization of EIoU as the loss function, addition of an attention module, and incorporation of the detection head have collectively led to an increase in the mAP of the YOLOv8 model for yellow-peach detection from 79.9% to 84.1%, thereby achieving superior performance.

### 4.3. Comparison of Different Networks

In order to demonstrate the advantages of the EMA-YOLO model, we conducted a performance comparison with other common object-detection-algorithm models. The classical object-detection network SSD [29], which is based on object regression (with vgg-16 backbone network), as well as Objectbox [30] and the YOLO [31] series, including YOLOv7-Peach [15], were included for comparison. The results of the comparison are presented in Table 2.

The table illustrates that the EMA-YOLO model demonstrates superior precision and recall rates compared to other models, with a mAP of 84.1%. Specifically, the EMA-YOLO model shows a 1.5% improvement in precision compared to YOLOv8, along with a 3.6% increase in recall rate. This indicates a reduced rate of missed detections and enhanced overall accuracy. The orchard environment presents complexities such as background leaves that are similar in color to immature peaches, occlusions from dense fruit distribution, and numerous small targets. However, the improved network model addresses these issues by incorporating an EMA attention mechanism and additional small-target detection head. As a result, there are significant improvements in recall rate. While Single Shot Multibox Detector (SSD) boasts high precision with this dataset, its low recall rate limits its mAP (only 54.0%). On the other hand, ObjectBox—a recently developed anchor-free object-detection network—achieves a precision of 83.8% and a recall of 61.4%, yet falls short in terms of its mAP, at only 69.9%.

In summary, the EMA-YOLO model successfully strikes a balance between high precision and recall rates that align with project requirements.

### 4.4. Comparison at Different Shooting Distances

Capturing images in a natural environment makes it nearly impossible to maintain constant camera angles and shooting distances. Thus, it is necessary to ensure that objects of different sizes are effectively detected. However, manual methods may overlook some small yellow peaches. Therefore, it is essential to verify the model’s performance by detecting yellow-peach images at different shooting distances. Figure 9 shows the detection effectiveness of YOLOv8 and EMA-YOLO.

Table 3 summarizes the result of the detection comparison. Compared with the aforementioned experimental, it is evident that there is minimal disparity between the YOLOv8 model and EMA-YOLO model at short distances. In scenarios (a) and (b), EMA-YOLO achieved the perfect result, with no peaches missed, while YOLOv8 missed only one peach in both scenarios. At a moderate distance, YOLOv8 missed three peaches and EMA-YOLO missed two peaches in scenario (a). In scenario (b), YOLOv8 missed three yellow peaches compared to EMA-YOLO’s single miss. However, for long-distance image detection, compared to detection at short and moderate distances, there are more severe cases of missed detections: in scenario (a), seven yellow peaches were overlooked; in scenario (b), ten were omitted by YOLOv8, whereas only one or two were overlooked by EMA-YOLO. In summary, despite some missed detections by EMA-YOLO, its rate is lower than that of the YOLOv8 model, demonstrating its superior performance in the detection of unmatured small yellow peaches.

### 4.5. Comparison of Different Light Intensities

When working in the orchard, it is important to consider weather conditions. For instance, images captured under strong light typically display higher contrast and more pronounced shadows and highlights of objects, whereas those images acquired under low light may show reduced contrast and potential blurring of object details. Additionally, images captured in low light may also suffer from noise or blur, impacting the clarity of details, while high light allows for clearer detail capture.

To assess the robustness of the EMA-YOLO model, we conducted tests under varying light intensities. The results are depicted in Figure 10.

Table 4 summarizes the ground truth and numbers of peaches detected by various models. It is apparent that while YOLOv8 missed only two yellow peach at most during detections conducted under strong or moderate light intensities, it failed to detect as many as seven or even ten samples when subjected to low-light conditions. In contrast, EMA-YOLO demonstrated significantly better performance, missing only three instances at most in all scenarios. These findings highlight the superior detection capabilities of EMA-YOLO.

Our analysis also revealed that images captured under weak illumination often suffer from noise or blurring effects, consequently impacting accurate target identification. By integrating an EMA attention-mechanism module into its backbone network along with a dedicated small-target detection head component, EMA-YOLO effectively amplifies feature extraction across diverse information types while prioritizing crucial features over interfering ones. Furthermore, this combination minimizes original data loss due to network transmission, thereby enhancing focus on information relating to smaller targets.

### 4.6. Comparison of Different Densities

During the shooting process, it was observed that yellow peaches tend to have a dense growth distribution in the natural environment. This results in occlusion between the fruits and also between the fruits and leaves. Such occlusion hinders accurate extraction of certain characteristics of the yellow-peach fruits during target detection, leading to missed identification of partially occluded peaches. Therefore, in order to validate the superiority of the EMA-YOLO model, it is essential to compare its detection capability with that of the YOLOv8 model for yellow-peach targets at different densities, as shown in Figure 11.

The results are summarized in Table 5. At sparse densities, neither EMA-YOLO nor YOLOv8 missed any peaches in scenario (a). In scenario (b), the YOLOv8 model missed two objects, while the EMA-YOLO model did not miss any. Thus, the EMA-YOLO model is better than YOLOv8 in this case. When dealing with a moderately dense yellow-peach distribution, YOLOv8 missed five and six yellow peaches in two different figures. On the other hand, EMA-YOLO missed only two yellow peaches in each figures respectively.

In the case of extremely dense distributions of yellow peaches, there is a substantial difference between YOLOv8 and EMA-YOLO. In scenario (a), YOLOv8 detected only 145 yellow peaches; although EMA-YOLO also missed five yellow peaches, there was a difference of 17 yellow peaches compared with YOLOv8’s results. In scenario (b), YOLOv8 missed 14 yellow peaches, 10 more than EMA-YOLO. These results show that in the case of a very dense distribution of yellow peaches, YOLOv8 has a serious missed-detection problem due to the severe occlusion and the small size of yellow peaches. This result also indicates that EMA-YOLO has better detection capabilities.

Furthermore, the convolutional neural network experiences information loss after multiple passes, leading to occlusion in dense distributions and resulting in inaccurate or missed detection. The EMA attention-mechanism module enhances the ability of the EMA-YOLO model to extract information from occluded yellow peaches, prioritizing the retention of information that may be lost during layer-wise transmission. This improvement allows for more accurate detection of severely occluded yellow-peach features in densely distributed areas. Consequently, it significantly improves detection performance under these conditions.

### 4.7. Comparison of Computational Load

Model parameter number (Params) is a metric used to evaluate the spatial complexity and scale of the model, so low parameter number is an important index indicating a lightweight model. Furthermore, model computation (GFLOPs) is the number of floating-point operations performed by a model in one forward propagation, usually in billions of floating-point operations per second. The model computing power is used to evaluate the computing-resource consumption of the model. Lower computing-power requirements are more applicable to devices or scenarios with limited memory or computing power. Table 6 summarizes the Params and GFLOPs of our model.

By combining the EMA with the small-target detection head, the network can detect more small targets and improve the accuracy of its results. However, the introduction of the EMA module and the small-target detection head inevitably led to an increase in the parameter number and number of model calculations, and the introduction of loss function cannot make the model lightweight. Although a certain amount of computation is sacrificed, in the visualization results, the model shows obvious improvement in small-target detection and performance under conditions of severe occlusion. In the context of increasingly abundant storage and computing resources, the accuracy of the model should be considered first.

### 4.8. Discussion

After the EMA attention-mechanism module was integrated and a small-target detection head was added to the YOLOv8 model, the experimental comparison clearly demonstrated improved small-target detection, as well as performance enhancement under different lighting and intensity levels. This led to a reduction in missed detections and an overall improvement in the accuracy of the object-detection algorithm. Although the introduction of the EMA module and the addition of the small-target detection head inevitably increased the number of parameters and model calculations, the model has obviously improved performance for small-target detection and under conditions of severe occlusion. The Grad-CAM method [32] is commonly used to improve the interpretability of neural network models by generating heat maps based on weight features extracted from different layers.

As shown in Figure 12, it is evident that the improved EMA-YOLO model exhibits more hierarchical red coloration for detected targets compared to the original YOLOv8 model, particularly when dealing with small yellow-peach targets against similar-colored backgrounds.

According to the above three groups of comparative experiments (Figure 9, Figure 10 and Figure 11), it can be seen that for yellow-peach images taken at short and moderate distances, yellow-peach images taken at strong and moderate light intensities, and yellow-peach images with sparse distributions, the detection results of the EMA-YOLO model include fewer missing targets. Therefore, the EMA-YOLO model has certain advantages in the detection of yellow peaches in orchards and this can basically meet the needs of agricultural detection. However, when the distribution of yellow peaches is dense and the target of yellow peaches is small, the detection capacity of EMA-YOLO for yellow peaches is not ideal. There are more missed detections (refer to Figure 11 and Table 5), which may be due to the fact that there is less feature information for occluded yellow peaches in the dense distribution with low resolution, resulting in the failure to extract some features. In view of this, the feature extraction of the input picture information should be strengthened in the subsequent research process to reduce the loss of information caused by the increase in network layers and thus further improve the accuracy.

A method named YOLOv7-Peach, proposed in reference [15], has a precision rate of 79.3%, a recall rate of 73%, and an average accuracy of 80.4%. Compared with YOLOv7-Peach, our method has a higher recall rate, is more suitable for counting function, and reduces the number of missed detections. Compared with YOLOv7, our method has a 4.3% higher precision, a 1.4% higher recall rate, and a 3.7% higher mAP rate on the same data set. Reference [15] does not give the relevant computational load index, so it is not possible to make further comparisons in terms of the “lightweight” aspect.

## 5. Application of Our Method

To validate the practical applicability of the method proposed in this paper, an application system will be developed based on the cloud and mobile terminals. The system is primarily built on Android technology. Users can take photos or upload local images via the mobile terminal; the images are processed in the cloud, and the results are fed back to the terminal. The specific test model used is a Xiaomi device running Android 12.0. The test results interface is illustrated in Figure 13.

## 6. Conclusions

This paper addressed the problem of the detection of yellow peaches in the natural environment and designed the EMA-YOLO model based on YOLOv8. The model utilizes data-augmentation technology to expand the sample space and increase sample diversity. It introduces the EMA attention-mechanism module to encode global information and further aggregate pixel-level features through dimensional interaction, thereby reducing missed and false detections of small targets in densely occluded environments and improving overall accuracy. Additionally, a small-target detection head is incorporated to enhance the ability of the model to detect small targets, thus improving the detection rate. Furthermore, replacing the loss function with EIoU reduces instances of missing and false detections of target small yellow peaches in dense scenarios. The model demonstrates certain advantages in detecting yellow peaches under various environmental conditions in peach orchards, providing valuable insights for the accurate detection of small targets against similar background colors and offering technical support for intelligent yield-estimation management of yellow peaches.

## Figures and Tables

**Figure 1 sensors-24-03783-f001:**
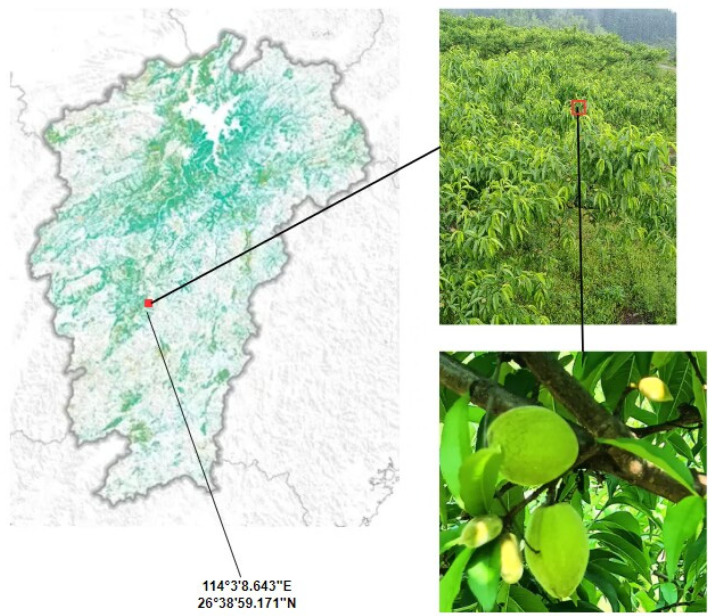
The location of the plantation base.

**Figure 2 sensors-24-03783-f002:**
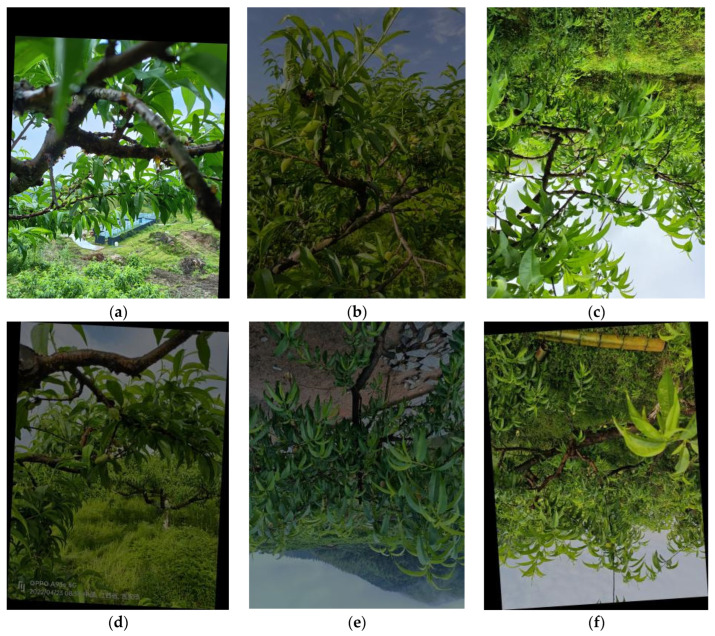
Examples of images from the yellow-peach dataset. (**a**) random rotation (**b**) change in brightness (**c**) flip. (**d**) random rotation and change in brightness (**e**) flipping and change in brightness (**f**) flipping and random rotation.

**Figure 3 sensors-24-03783-f003:**
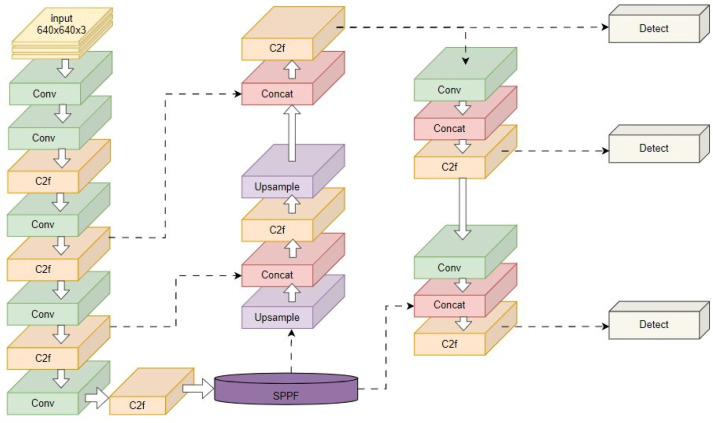
YOLOv8 network architecture.

**Figure 4 sensors-24-03783-f004:**
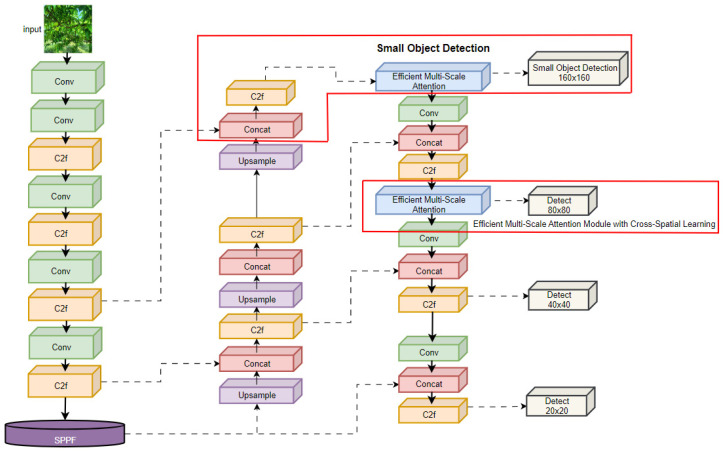
Network structure of the EMA-YOLO (red boxes represent the improvements).

**Figure 5 sensors-24-03783-f005:**
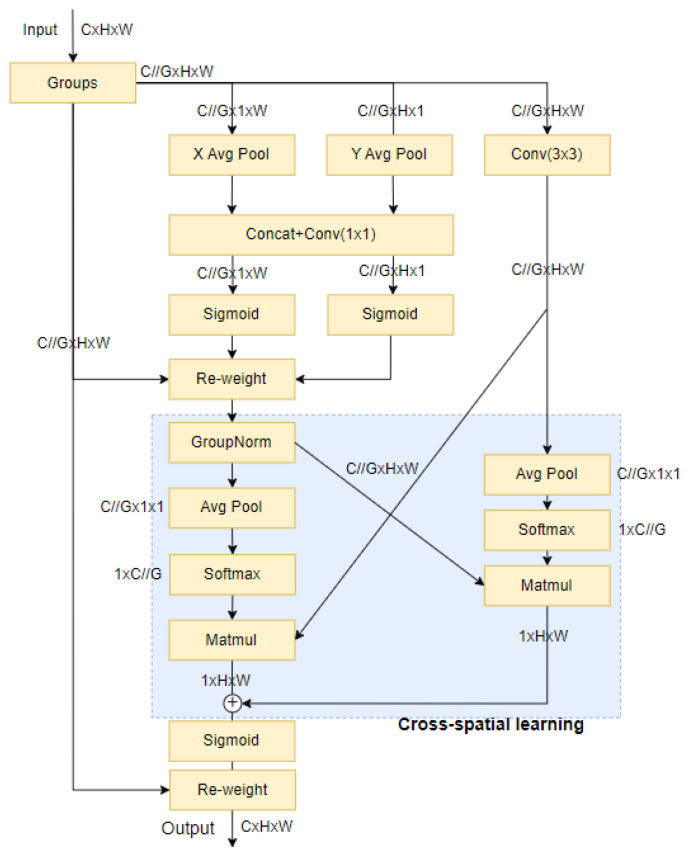
Structure diagram of the EMA attention-mechanism module.

**Figure 6 sensors-24-03783-f006:**
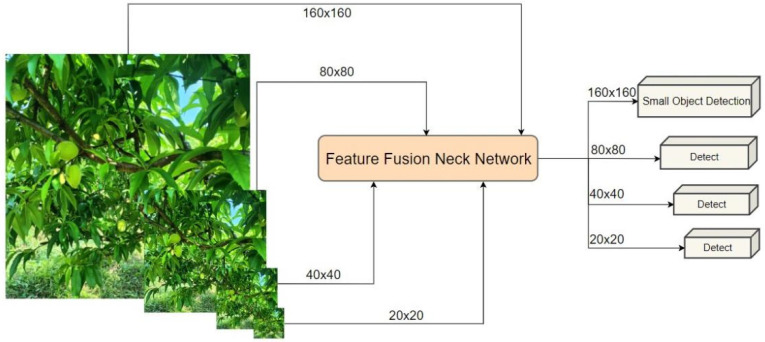
Improved detection layer.

**Figure 7 sensors-24-03783-f007:**
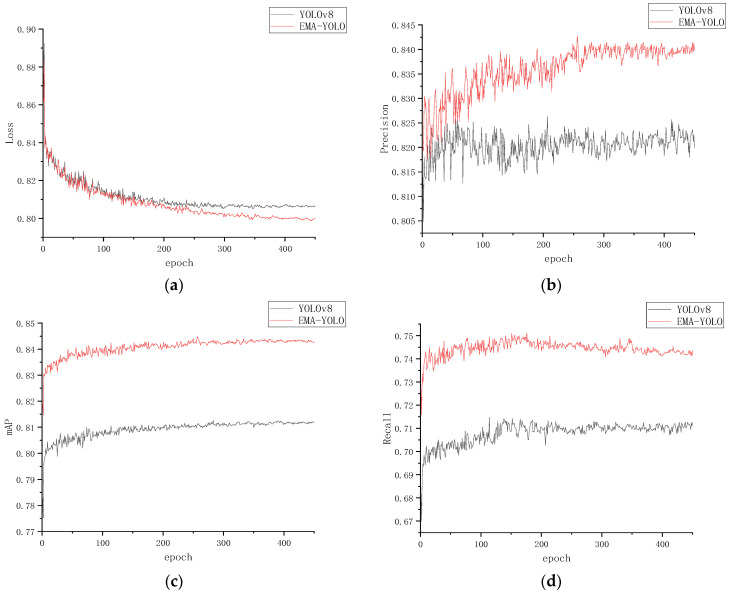
The curves of the training process. (**a**) Loss curve; (**b**) precision curve; (**c**) mAP curve; (**d**) recall curve.

**Figure 8 sensors-24-03783-f008:**
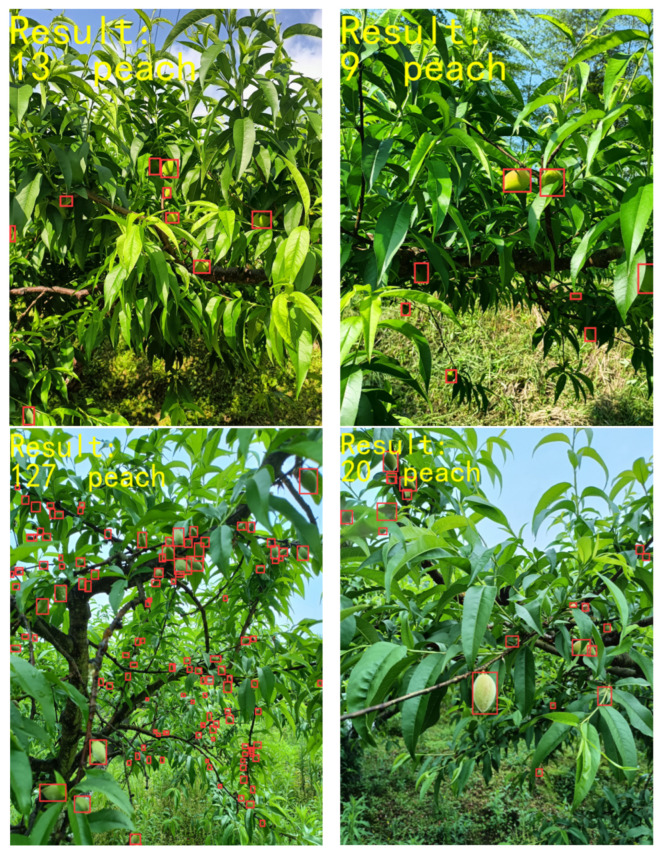
Partial test results.(☐ in the images represents the detected yellow peach).

**Figure 9 sensors-24-03783-f009:**
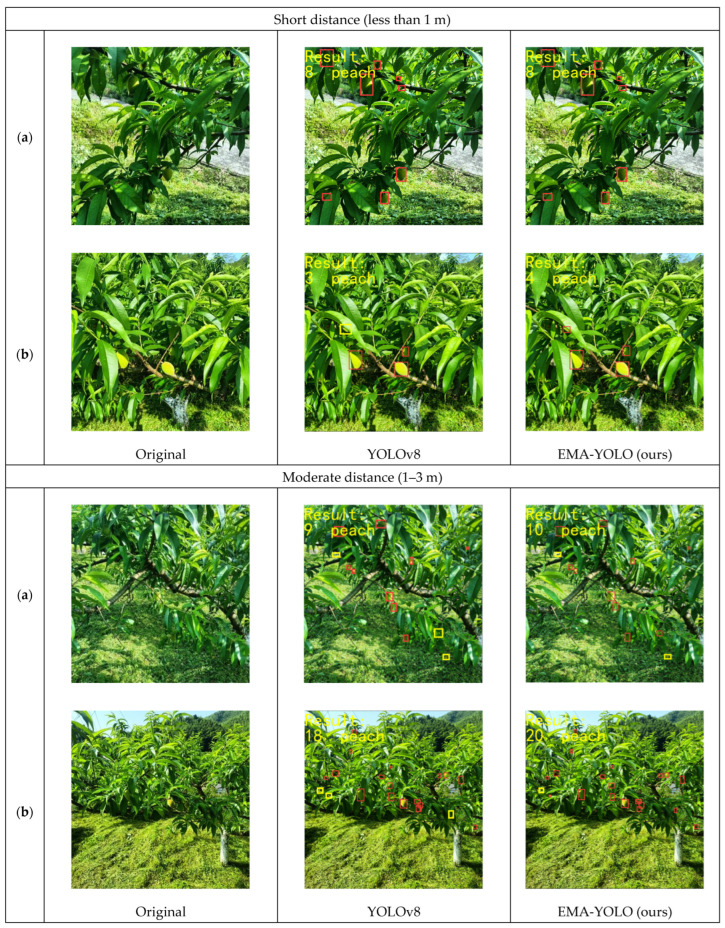
Detection at different shooting distances. (**a**) and (**b**) are two different images under the same environmental conditions. ☐ in the images represents the detected yellow peach, and ☐ represents the missed yellow peach.

**Figure 10 sensors-24-03783-f010:**
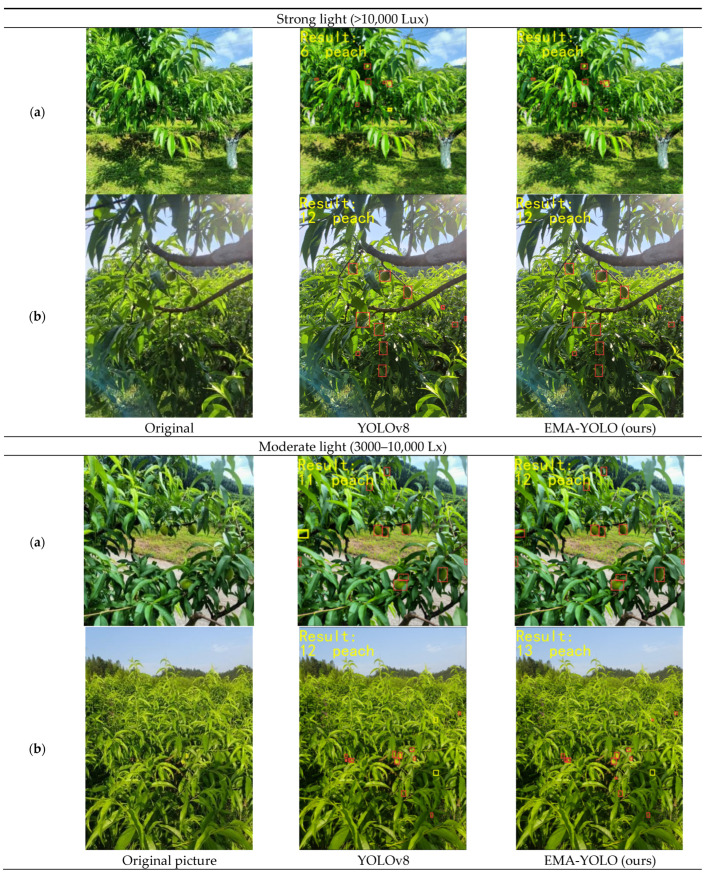
Detection under different light intensities. (**a**) and (**b**) are two different images under the same environmental conditions. ☐ in the images represents the detected yellow peach, and ☐ represents the missed yellow peach.

**Figure 11 sensors-24-03783-f011:**
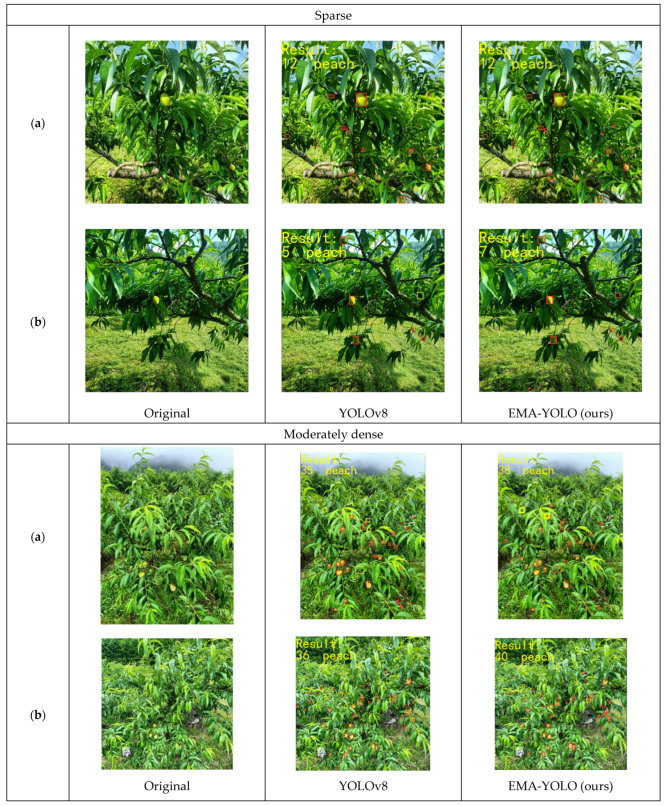
Detection at different densities. (**a**) and (**b**) are two different images under the same environmental conditions. ☐ in the images represents the detected yellow peach, and ☐ represents the missed yellow peach.

**Figure 12 sensors-24-03783-f012:**
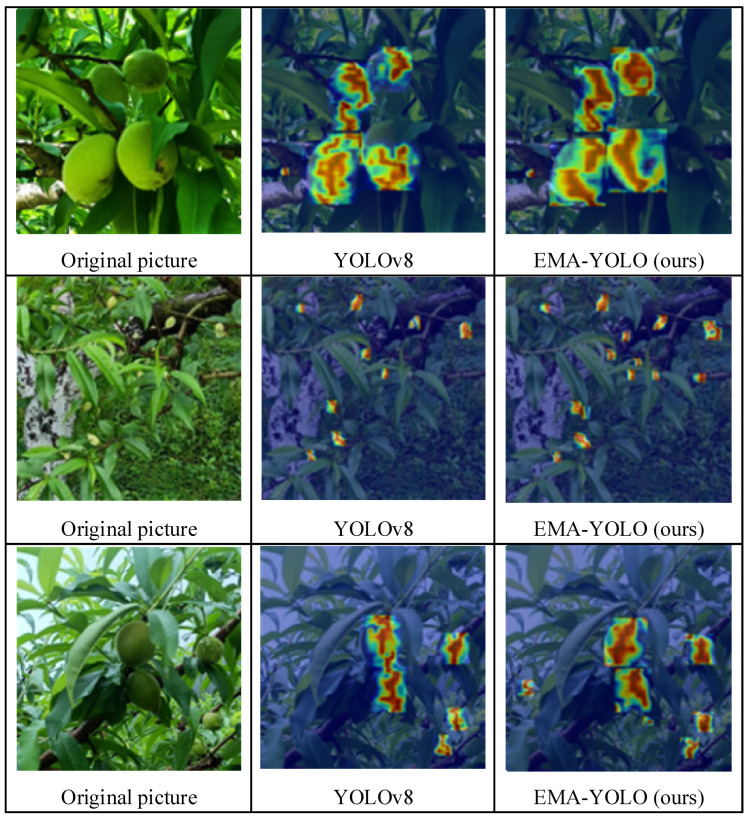
Visualization results.

**Figure 13 sensors-24-03783-f013:**
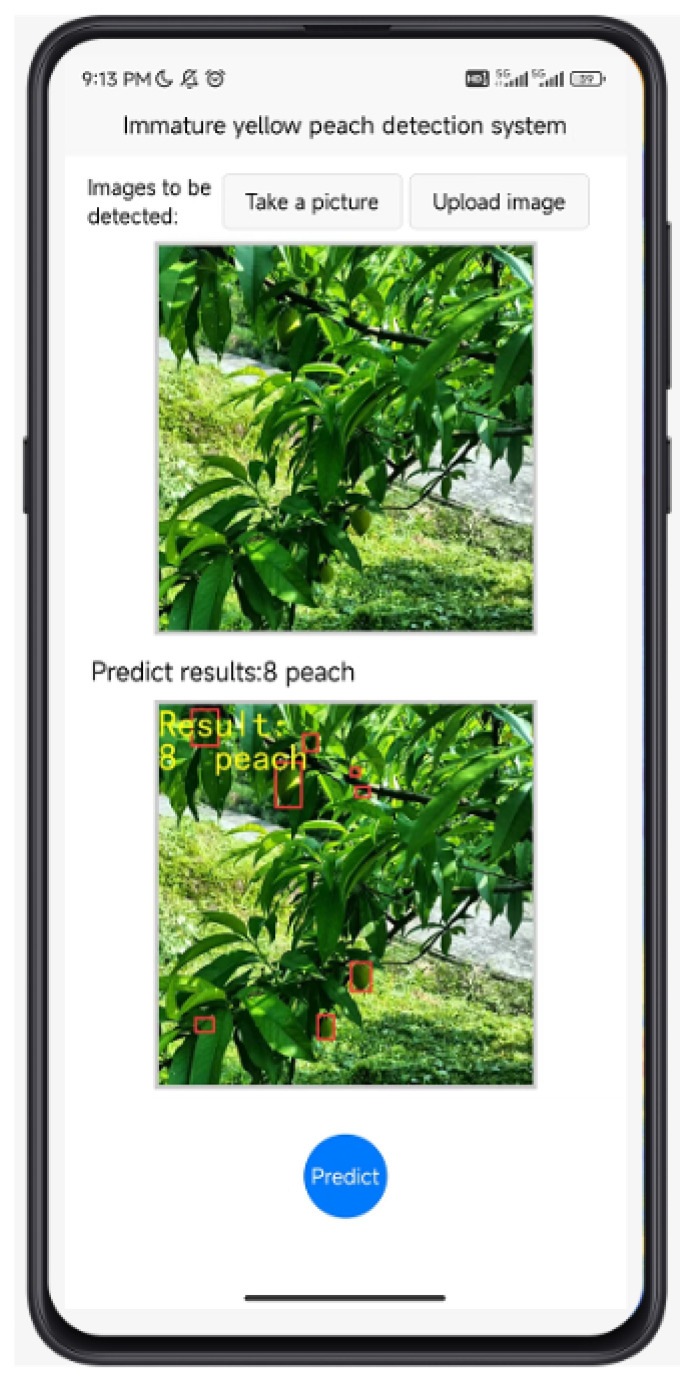
Application interface. (☐ in the image represents the detected yellow peach).

**Table 1 sensors-24-03783-t001:** Ablation experiments.

Experiment	Data Augmentation	EMA	Small Object Detection	EIoU	mAP
A					79.9%
B	√				81.5%
C	√	√			82.6%
D	√	√	√		83.5%
E	√	√	√	√	84.1%

Note: √ means we used this module.

**Table 2 sensors-24-03783-t002:** Network-comparison table.

Target-Detection Model	mAP	P	R	F1	mAP@.5:95
SDD-VGG	0.540	0.933	0.170	0.288	0.225
YOLOv3	0.739	0.820	0.665	0.734	0.370
YOLOv4s	0.749	0.813	0.650	0.722	0.364
YOLOv5n	0.685	0.787	0.610	0.687	0.312
YOLOv7n	0.769	0.793	0.697	0.742	0.371
YOLOv7-Peach	0.804	0.793	0.730	0.760	0.396
YOLOv8n	0.799	0.821	0.708	0.760	0.418
ObjectBox	0.699	0.838	0.614	0.709	0.339
EMA-YOLO(ours)	0.841	0.836	0.744	0.787	0.447

**Table 3 sensors-24-03783-t003:** Detection results at different shooting distance.

Models	Different Conditions	Scenario	Ground Truth	Predicted	Missed
YOLOv8	Short distance	(a)	8	8	0
		(b)	4	3	1
	Moderate distance	(a)	12	9	3
		(b)	21	18	3
	Long distance	(a)	21	14	7
		(b)	43	33	10
EMA-YOLO(ours)	Short distance	(a)	8	8	0
		(b)	4	4	0
	Moderate distance	(a)	12	10	2
		(b)	21	20	1
	Long distance	(a)	21	20	1
		(b)	43	41	2

**Table 4 sensors-24-03783-t004:** Detection results under different light intensities.

Models	Different Conditions	Scenario	Ground Truth	Predicted	Missed
YOLOv8	Strong light	(a)	7	6	1
		(b)	12	12	0
	Moderate light	(a)	12	11	1
		(b)	14	12	2
	Dim light	(a)	35	28	7
		(b)	33	23	10
EMA-YOLO(ours)	Strong light	(a)	7	7	0
		(b)	12	12	0
	Moderate light	(a)	12	12	0
		(b)	14	13	1
	Dim light	(a)	35	32	3
		(b)	33	31	2

**Table 5 sensors-24-03783-t005:** Detection results under conditions of different densities.

Models	Different Conditions	Scenario	Ground Truth	Predicted	Missed
YOLOv8	Sparse	(a)	12	12	0
		(b)	7	5	2
	Moderately dense	(a)	40	35	5
		(b)	42	36	6
	Extremely dense	(a)	167	145	22
		(b)	131	117	14
EMA-YOLO(ours)	Sparse	(a)	12	12	0
		(b)	7	7	0
	Moderately dense	(a)	40	38	2
		(b)	42	40	2
	Extremely dense	(a)	167	162	5
		(b)	131	127	4

**Table 6 sensors-24-03783-t006:** Computational load comparison table.

Model and Module	Params/M	FLOPs/G
YOLOv8n(X)	12.04	8.9
X+P2 Detect(Y)	12.79	17.4
Y+EMA(Z)	12.80	17.6
Z+EIoU	12.80	17.6

## Data Availability

Given that the data used in this study were self-collected, the dataset is being further improved. Thus, the dataset is unavailable at present.

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
