# Peer review of "EMA-YOLO: A Novel Target-Detection Algorithm for Immature Yellow Peach Based on YOLOv8"

_sensors, 2024, doi:10.3390/s24123783_

Round 1

Reviewer 1 Report

Comments and Suggestions for Authors

The manuscript fits within the scope of this section of the journal. From a general point of view, the description of the EMA-YOLO methodology and experiments is clear and easy to understand. The experimental section explores the features and advantages of the proposed model from several perspectives. However, the article needs further polishing to highlight the novelty of the application area. In order to improve the quality of the article, it needs to be revised according to the following comments:

 1. Introduction: What are the features, and innovations in the field of this paper (yellow peach detection)? To better compare with previous work and to highlight the uniqueness of this paper's field, add a summary of detection studies related to yellow peach in recent years (if there are fewer studies in this field, this should also be noted in the manuscript). 

2. The number of samples in the dataset is not very large, how to ensure that the model will not be overfitted? Can data enhancement solve the related problems? Also, please give the number of samples after performing data augmentation.

 3. Line 202 mentions "this improvement slightly increases computational load", which needs more analysis in the theoretical or experimental discussion section.

 4. Section 4.1 and 4.2:Experimental result and Ablation experiment. Comparison and discussion of the experimental results are not thorough and adequate and should be analyzed in conjunction with theoretical models. In addition, as "computational load" was mentioned above, and the model is also relevant to UAV deployment, it is suggested that the experimental section should preferably provide parameters results and discussions related to the size and computational complexity of the model. 

5. Discussion section: additional emphasis on the differences and advantages and disadvantages with other similar studies. Please check Figure 14 and Table 6 for duplications with the previous experimental results, and any relevant discussion can be moved to the Experimental section.

Comments on the Quality of English Language

The manuscript contains a number of detailed issues that need to be revised, thoroughly check the entire article for any further issues with grammar spelling, etc. Some of the issues suggested for revision are listed below:

-Table 1 and 2: mAp->mAP? 

-line 9: is->are 

-line 12: angle->angles 

-line 32: famers->farmers 

-line 38: estimate-> estimating? 

-line 101: image->images 

-line 136: “In the Head section, it employs the…”->The Head section employs the”?

-line 158: tree->the tree, it->them? 

-line 163: resultful method->a resultful method? 

-line 192: resulting in potential loss-> resulting in a potential loss? 

-line 202: computational load-> the computational load? 

-line 337: abalance-> a balance? 

-line 374: model-> models 

Please carefully check the full manuscript for details similar to those mentioned above to ensure fluency and correctness of expression.

Reviewer 2 Report

Comments and Suggestions for Authors

This study proposed an improved target detection model based on YOLOv8 for detecting small yellow peaches. This study has a certain role in promoting the automatic detection of yellow peach. However, there are still certain problems with the article that need to be improved. Please see PDF for specific comments.

Comments on the Quality of English Language

Extensive editing of English language required.

Author Response

Please see the attatchment.

Reviewer 3 Report

Comments and Suggestions for Authors

In this paper, the authors proposed a yellow peach detection method based on improved YOLOv8. Overall, the paper is well-written and organized. While the novelties of the method are minor given the dataset is not accessible. In addition, the statistics (like Table 2 rather than Table 3) of the different distances are not given. 5-fold Cross-validation is suggested to be included to remove the randomness of the dataset partition. The app demo is good, but it is highly recommended to change the interface to English.

Comments on the Quality of English Language

None

Author Response

Please see the attatchment.

Round 2

Reviewer 2 Report

Comments and Suggestions for Authors

I'm glad the authors took my advice and the article has been greatly improved from the first edition. I think the article is acceptable with minor revisions.

The specific comments are as follows:

1. the evaluation indicators should retain the same number of decimal places. For example, Table 2.

2. Table 1 figures are not aligned.

3. Figures 7, 8, 9 and 10 can be combined into one picture.

4. References should be placed before the period.

5. The format of the figure notes in Figure 2 is confusing.

Reviewer 3 Report

Comments and Suggestions for Authors

NA

Author Response

Dear Editors and Reviewers:

We appreciate for Editors/Reviewers’ warm work earnestly, and hope the correction will meet with approval. Thank you again for your positive comments and valuable suggestions to improve the quality of our manuscript.